# Emerging CAR T Cell Strategies for the Treatment of AML

**DOI:** 10.3390/cancers14051241

**Published:** 2022-02-27

**Authors:** Paresh Vishwasrao, Gongbo Li, Justin C. Boucher, D. Lynne Smith, Susanta K. Hui

**Affiliations:** 1Department of Radiation Oncology, City of Hope Medical Center, Duarte, CA 91010, USA; shui@coh.org; 2Department of Hematology, Academic Medical Center, University of Amsterdam, 1105 AZ Amsterdam, The Netherlands; 3Department of Neurological Surgery, Feinberg School of Medicine, Northwestern University, Chicago, IL 60611, USA; gongbo.li@northwestern.edu; 4Department of Blood and Marrow Transplant and Cellular Immunotherapy, Division of Clinical Science, H. Lee Moffitt Cancer Center, Tampa, FL 33612, USA; justin.boucher@moffitt.org; 5Department of Hematology, City of Hope Medical Center, Duarte, CA 91010, USA; lysmith@coh.org

**Keywords:** acute myeloid leukemia (AML), chimeric antigen receptor (CAR)-T cells, co-stimulatory domains, preclinical model, single-chain variable fragment (scFv)

## Abstract

**Simple Summary:**

Chimeric antigen receptors (CARs) targeting CD19 have emerged as a new treatment for hematological malignancies. As a “living therapy”, CARs can precisely target and eliminate tumors while proliferating inside the patient’s body. Various preclinical and clinical studies are ongoing to identify potential CAR-T cell targets for acute myeloid leukemia (AML). We shed light on the continuing efforts of CAR development to overcome tumor escape, exhaustion, and toxicities. Furthermore, we summarize the recent progress of a range of putative targets exploring this unmet need to treat AML. Lastly, we discuss the advances in preclinical models that built the foundation for ongoing clinical trials.

**Abstract:**

Engineered T cells expressing chimeric antigen receptors (CARs) on their cell surface can redirect antigen specificity. This ability makes CARs one of the most promising cancer therapeutic agents. CAR-T cells for treating patients with B cell hematological malignancies have shown impressive results. Clinical manifestation has yielded several trials, so far five CAR-T cell therapies have received US Food and Drug Administration (FDA) approval. However, emerging clinical data and recent findings have identified some immune-related toxicities due to CAR-T cell therapy. Given the outcome and utilization of the same proof of concept, further investigation in other hematological malignancies, such as leukemias, is warranted. This review discusses the previous findings from the pre-clinical and human experience with CAR-T cell therapy. Additionally, we describe recent developments of novel targets for adoptive immunotherapy. Here we present some of the early findings from the pre-clinical studies of CAR-T cell modification through advances in genetic engineering, gene editing, cellular programming, and formats of synthetic biology, along with the ongoing efforts to restore the function of exhausted CAR-T cells through epigenetic remodeling. We aim to shed light on the new targets focusing on acute myeloid leukemia (AML).

## 1. Introduction

Adoptive immunotherapies, which have shown promise in treating hematological malignancies, can target acute myeloid leukemia (AML) through distinct and complementary pathways. Adoptive T cell immunotherapy may be particularly potent due to the longevity and strong cytotoxic activity of transferred T cells. One such adoptive immunotherapy is chimeric antigen receptors (CARs), which are single-molecule recombinant antigen receptors that can redirect T cell specificity and enhance anti-tumor potency. These synthetic hybrid receptors are composed of a cell-surface ligand molecule fused to signaling domains assembled to redirect T cell function. The extracellular ligand molecule can be derived from a single-chain variable fragment (scFv) derived from a monoclonal antibody or an antigen-binding fragment (Fab). This genetic modification provides a T cell with an alternative antigen specificity. Upon binding to a specific antigen, the CAR initiates signaling and the activation of the T cell leading to target cell killing. Many tumors down-regulate the expression of the major histocompatibility complex (MHC) to avoid the T cell anti-tumor effects. This resistance is avoided due to the functional nature of the MHC-independent mode of action of CAR-T cells. CAR intracellular domains can incorporate both T cell receptor signaling by CD3ζ and co-stimulatory signalings such as CD28 and/or 4-1BB, which have proven particularly effective in simultaneously promoting both T cell survival and effective target killing.

A defined regimen has been approved by the US Food and Drug Administration (FDA) for the treatment of certain B cell malignancies by CAR-T cell therapy. The regimen begins with collecting blood from the patients in the clinic. Patients are conditioned before CAR-T cell infusion in most cases and then undergo leukapheresis, followed by T cell isolation from their peripheral blood mononuclear cell (PBMC) fraction. Furthermore, these collected T cells undergo ex vivo activation with either beads or exogenous supplementation of the cytokines. A gene for CAR expression is then introduced either through mRNA, lentiviral, or retroviral methods into these autologously expanded T cells to redirect the cells to recognize tumors. Following the expansion of these CAR-T cells, they are introduced into the patient and monitored with flow cytometry. Upon activation, they can survey for the presence of tumor cells constantly. 

Preclinical efforts of CAR-T cells and their modifications have led to several clinical trials. Currently, there are five FDA-approved CAR-T cell therapies: Tisagenlecleuel [1,2] (KYMRIAH; Novartis), Axicabtagene ciloleucel [3,4] (YESCARTA, Kite Pharma, Gilead Science), Brexucabtagene autoleucel (TECARTUS; Kite Pharma), l Isocabtagene maraleucel (BREYANZI; Juno Therapeutics, Inc., a Bristol-Myers Squibb Company) and Idecabtagene vicleucel (ABECMA; Celgene Corporation, a Bristol-Myers Squibb Company). The majority of the approved CAR-T cells therapies target CD19, a B-cell-specific antigen that has shown tremendous results in the clinical trials against acute lymphoblastic leukemia (ALL) and non-Hodgkin lymphoma [5]. 83% of the people treated with the first approved CAR-T drug, “Kymriah” from Novartis, achieved complete remission within 3 months. The therapeutic potentials of adoptive transfer T cells expressing CARs have shown clinical success against advanced B cell malignancies [6,7] and relapsed and refractory (r/r) leukemia [6,8]. The development of CARs has not been limited to hematological cancer but is extensively expanding to solid tumors [9]. 

In the past decade, second-generation CARs targeting CD19 were found to be the most efficacious adoptive T cell therapy to date. Compared to the first generation, second-generation CARs endowed higher potential of anti-tumor and long-lasting responses in high-grade B cell lymphoma in adults [10,11,12] and in B cell acute lymphoblastic leukemia (B-ALL) in both children and adults who are refractory to all standard therapies [13,14,15,16]. In general, B-cell-related hematological malignancies were initially targeted with CD19 and CD20, which are differentially expressed on B cell lymphocytes. The implementation of CD19-CAR to target leukemia was initially developed from preclinical studies published after immediate succession by several groups (Micheal Sadelain, Steven Rosenberg, Stephen J. Forman, and Carl June). Among all the CARs, CD19 CAR-T cell therapy for acute lymphoblastic leukemia (ALL) has shown the most promising results (>85%) as the CARs that respond to the pan-B-cell marker CD19 have shown robust efficiency. Though the anti-tumor cytotoxicity was directed towards targeting malignant B cells expressing CD19, normal healthy B cells were also targeted. The off-tumor toxicity led to B-cell aplasia, a condition that is clinically managed with the prophylactic infusion of γ-globulin [17]. The addition of the costimulatory domains CD28, and 4-1BB (second-generation CARs), to CAR design not only elevated the anti-tumor activity significantly but prolonged in vivo survival efficiency of the CAR-T cells. There was unhindered accessibility of the target antigen in circulating tumor cells, without any major immunosuppressive factors, such as CD95/FasL and cytokines such as interleukin (IL)-10 and transforming growth factor (TGF-β). All of these conditions together made CD19 a more suitable candidate for CAR-T cell therapy, yet the clinical results reported some disadvantages during clinical trials. Emerging new evidence has identified critical technological and biological roadblocks. A significant concern is the lack of patient T cells, which are required to produce the CAR-T cells. With these challenges, scientists are gearing towards simplifying the feasibility and logistics of CAR-T cell generation with the concept of “universal CAR-T cells”, which would function as “off-the-shelf” CAR-T cells [18]. This review elaborates on the current understanding of the biology behind CAR design, signaling, expression and functionality, and experimental CAR designs targeting AML.

## 2. Architecture of CARs

### 2.1. Design and Composition

CARs are synthetic proteins that elicit anti-tumor activity after binding to their target antigen. They consist of an extracellular antigen-binding domain, typically a single-chain variable factor (scFv), extracellular spacer, a transmembrane domain, and intracellular domains for T cell activation. A standard intracellular domain is composed of a CD3ζ activation signal that is linked to a co-stimulatory proliferation signal, both of which are activated upon antigen binding, allowing the T cell to essentially act as a “*killing machine*” [19]. Conceptually, a perfect CAR target for leukemia should consist of an expression at high density in all tumor cells, including all the heterogeneous clones that include leukemic stem cells (LSCs), in addition to avoiding targeting normal tissue (any organs) or hematopoietic stem progenitor cells (HSPCs).

First-generation CARs were originally co-created by Zelig Eshhar in 1989. T cell lymphocytes were transfected with a splice variant of a heavy and light chain variable region of a monoclonal antibody (mAb) along with a constant portion of TCR [20]. These first-generation CARs are composed of a single intracellular domain, mostly CD3ζ, which triggered T cell activation. Due to their limited signaling capability, the first-generation CARs were unable to prime resting T cells or sustain lasting T cell response or cytokine release. Second- and third-generation CARs, respectively, have an additional one or two intracellular costimulatory signaling domains. These additional domains were incorporated to enhance T cell activation, persistence and proliferation [21]. These well-defined domains of CAR constructs have an advantage with each interchangeable structure imparting a functional diversity. Fourth-generation CAR T cells were engineered to include additional features to second-generation CARs, such as constitutive or inducible expression of the cytokine interleukin (such as IL-12) to support CAR activation. T cells transduced with these fourth-generation CARs are referred to as T cells redirected for universal cytokine-mediated killing (TRUCKs) [22,23]. Preclinical studies have reported greatly enhanced therapeutic benefits with reduced systemic toxicity [21]. Fifth- or next-generation CAR-T cells are currently being developed and are based on second-generation CARs but contain a truncated cytoplasmic IL-2 receptor β chain domain with a binding site for the transcription factor STAT3. This allows antigen activation of the receptor to simultaneously trigger TCR, costimulatory (CD28 domain) and cytokine (JAK-STAT3/5) signaling, all three of which are necessary for physiological T cell activation and proliferation. From all the pre-clinical in vitro and in vivo studies, as well as clinical outcomes from human trials, the field is still gaining new insights into a molecular understanding of T cell biology and it’s signaling effect through CAR design. Most of the preclinical in vivo findings stem from the adoptive transfer of human T cells into immunocompromised mouse models. Few clinical outcomes have reported some non-desirable “*on-target off-tumor*” effects. These adverse clinical outcomes have inspired pre-clinical researchers to revisit aspects of the CAR, including the rationale of the modulatory design, binding affinity and avidity, localization, sustained proliferation, activation, maintenance of the expanded immunophenotype, anti-tumor efficacy, sight and amount of injection, and, not to mention, an approach of combined immunotherapy. Figure 1 demonstrated the evolution of CAR design, improvising with every new generation.

#### 2.1.1. Single-Chain Variable Fragments (scFvs)

The scFv is the antigen-binding moiety of the CAR and is derived from the immunoglobulin heavy chain variable (V_H_) and Ig light chain variable (V_L_) domains. The receptor and/or ligand-binding domain needs to be functionally active once it is bound to the target antigen. Ligand affinity and avidity play an important role during antigen binding. CARs have also been constructed with antibody-derived binding components such as nanobodies [24] with some exceptions of natural binding partners of the target antigen.

#### 2.1.2. Extracellular Spacer

The most common spacers belong to the IgG Fc region, along with CD4 and CD8. Due to the risk presented by the interaction between Fc γ receptors (FcR_γ_) expressed by the NK cells, modification to the cognate Fc region is required. The addition or substitution of amino acids is done in the Fc portion to avoid the interaction with FcR_γ_ [25].

#### 2.1.3. Transmembrane Domain

The transmembrane region acts as a bridge between antigen-binding domains and intracellular domains spanning through the cell membrane. This structural region also serves to relay the signal from the outside to the inside of the T cell. These membrane-spanning domains are mostly from CD4, CD8, CD28, or CD3ζ.

#### 2.1.4. Costimulatory Domain

These domains are next to the transmembrane domain on the interior of the cell. These domains facilitate T cell proliferation, differentiation, activation, persistence, and contribute to cytokine production. CAR-T cells are then “charged” to take the course of action depending on which costimulatory domain has been included in their design. Native T cell receptors such as the CD28, inducible costimulator (ICOS) [26], OX40, 4-1BB have been used to gene-engineer in the CAR construct [27]. The most commonly used intracellular costimulatory domains are CD28 and 4-1BB. Significant improvements in clinical trials have been reported with the addition of costimulatory domains in first-generation CARs. The addition of 4-1BB to CARs has been reported to promote survival and persistence through upregulation of the anti-apoptotic factors, such as B-cell lymphoma-extra-large (Bcl-xl) [28]. Contrastingly, including the CD28 costimulatory domain has demonstrated increased activation of T cells in response to an antigen stimulus. CD28 costimulatory domain drive high T-cell activation, which leads to exhaustion and shortened persistence [29,30]. Currently, additional efforts are underway to generate third-generation CARs that combine multiple costimulatory domains [31,32]. Understanding the importance of signaling mechanisms, the effects of combined domains on transduced T cells, and equivalency over second-generation CARs will require more pre-clinical studies.

### 2.2. Activation Domain

The choice of activation domain included in first-generation CAR was CD3ζ or FcRγ. With the evolution in this field, CD3ζ has drawn more attention as the activation domain. Furthermore, the search for a better activation domain has led to the use of the CD3ζ activation domain in second-generation CAR, which has shown promising results in various clinical trials [3,7,21].

### 2.3. Signaling Mechanism

Different routes have been established to introduce the CARs in T cells, most commonly in CD8+ and/or CD4+ T cells. The viral transduction of CARs in T cells has achieved constitutive CAR expression [33,34], whereas the transfection of mRNA encoding transcripts has a clear advantage of transient expression with the durability of not more than a week [34,35]. CAR constructs with signaling domains derived from the cytoplasmic segments of CD3ζ, CD28, and 4-1BB are most commonly used in clinical CARs. Each domain of the CAR composition has been studied in depth to understand its functional application. CD28 and 4-1BB consist of positively charged, basic amino-acid residues in their cytoplasmic tails. Furthermore, while in a resting state, these residues have shown interaction with the negatively charged inner leaflet of the plasma membrane [36]. Initial studies of fusion proteins demonstrated that one of the cytoplasmic domains of the T cell receptor, CD3ζ, was sufficient to couple with the receptor-associated signal transduction. Once the receptor is bound to the cognate antigen, the cytoplasmic chain events are triggered [37]. One of the primary sites for the phosphorylation of CD3ζ required cytoplasmic immunoreceptor tyrosine-based activation motifs (ITAMs) [38,39]. Signal transduction requires the phosphorylation of these ITAMs by the Src family tyrosine kinases Lck and/or Fyn. The phosphorylated ITAMs then recruit and activate the cytoplasmic domain (Zap-70) [40] through the association between doubly phosphorylated ITAMs and tandem SH2 domains on Zap-70. Receptor triggering has been shown to activate intracellular chain events by initiating phosphatidylinositol and tyrosine kinase pathway together with calcium influx inhuman T cell leukemia [41,42]. Both the phosphorylation sites on CD3ζ and signaling-molecule binding sites on CD28 have an important role to play in aggregation, sequestration and regulation of signal transduction [37,43]. In summary, the configuration of the receptor chain, the position and location of the domains relative to the membrane, and the signaling molecules contribute to the initiation and activation of signal transduction [44]. The mechanism for the signal transduction and correlation of antigen binding on the extracellular region and the relaying of the signal through intracellular domains have been well studied. Signaling domains, which bind as immunoreceptors, have been reported to influence the cellular mechanism [21]. The impacts of biophysical and structural studies are currently under investigation. A recent review has summarized various elements and components that are required in CAR design [45]. 

## 3. CAR-T and Acute Myeloid Leukemia (AML)

In recent years, several tumor antigens, such as CD33, CD123, CLL-1, CD70, and TIM-3, have been explored as potential target antigens for AML treatment. A mono-antibody (mAb) therapy-directed targeting of these antigens has shown promising anti-tumor activity in animal models and clinical trials; yet, the overall therapeutic efficacies remain restricted. With the clinical success of CD19 CAR-T cell therapy on B-ALL for young and pediatric patients, CAR-T cells targeting these AML-associated antigens have been developed and have shown increasingly higher anti-tumor efficacy compared to mAb therapy.

### 3.1. Anti-FRβ-Specific CAR Therapy for AML

Powell et al. have previously reported that the folate receptor (FR) family of proteins is useful in T cell immunotherapy. FRα and FRβ are cell surface-bound proteins via glycosyl-phosphatidylinositol (GPI) linkages [46] that share almost 70% homology and a common mechanism of receptor endocytosis-mediated folate uptake. However, FRα and FRβ are differentially expressed in various tissues, with one being epithelial cell specific, while the other is expressed on myeloid-lineage hematopoietic cells, respectively. Both receptors are upregulated in a malignant setting [47]. FRβ is expressed on 70% of primary AML tumors, making it an attractive target for CAR-T cell immunotherapy. Interestingly, FRβ expression is upregulated on AML blasts when treated with all-trans retinoic acid (ATRA), a drug that is already approved by the FDA for subclass M3 AML [48]. FRα-directed CAR-T cells are already in a phase I trial on adoptive immunotherapy using gene-modified T cells for ovarian cancer [49]. The addition of co-stimulatory signaling through CD137 (4 1-BB) led to enhanced antitumor activity of CAR-engineered T cells through in vivo persistence and tumor localization [50]. Lynn et al. have generated and characterized human FRβ (m909)-specific CAR constructs containing the m909scFv, which recognizes human FRβ [10]. The validation of FRβ-specific CARs came through initial data where C30-FRβ-engineered cells confirmed the targeting of FRβ. The m909 CAR platform allows for efficient and specific targeting, demonstrated by in vivo culture assays and resulting in cytokine production, activation marker upregulation, proliferation, and target cell lysis when antigen levels are high. In the presence of human AML cells expressing FRβ, m909-CAR-T cells maintained specific activation in the presence of antigen. CAR activation is directly proportional to the expression of FRβ. With lowered FRβ expression, CAR activity was decreased as demonstrated by the reduced output of IFNγ and cytolysis. This is a concern with the potential for outgrowth of FRβ-low leukemic clones in m909 CAR-based therapy. However, the ATRA-specific upregulation of FRβ expression in FRβ+ AML cells and coculturing of m909 CAR-T cells with ATRA-pretreated AML exhibited higher IFNγ production and cytolytic activity. ATRA is known to induce differentiation of THP1 and HL60, as measured by greater cytokine production and co-stimulatory molecule expression [11,12]. ATRA did not impact FRβ expression in healthy HSCs or monocytes. This suggests that the ATRA-mediated induction of FRβ in AML could be applied without increasing the capacity for healthy tissue recognition by m909 CAR-T cells. Compared to ATRA alone, dual treatment with histone deacetylase (HDAC) inhibitors has been shown to further stimulate FRβ expression in AML in vitro [13]. With these data, an optimized combination of ATRA and FRβ-inducing agents could present an opportunity for additional augmentation of m909 CAR-T cell efficacy. FRβ expression is also found on normal myeloid lineage cells and can be induced in macrophage activation. This remains a potential target for on-target off-tumor toxicity by FRβ-specific CAR-T cells because the m909 CAR-T cells have not shown any lysis of peripheral blood monocytes from healthy donors.

### 3.2. Anti-CD33 CAR Therapy for AML

Almost 90% of leukemic cells in AML patients were expressed CD33 on myeloid cells [14,15]. CD33 is a myeloid-specific sialic-acid-binding receptor expressed on the majority of AML blasts and leukemic stem cells in patients, which was validated as an effective target in clinical trials [16]. Generation and implementation of monoclonal antibodies in preclinical and clinical trials have shown varying results. Lintuzumab is a humanized CD33-specific monoclonal antibody [17,51] that has shown promise in preclinical and early phases of clinical trials. However, there was no significant improvement in the response rate or survival compared with standard chemotherapy in subsequent trials [52,53]. Clinical trials with engineered CD33 (NCT 01690624) and several antibody-derived recombinant proteins targeting CD33 are in advanced preclinical development, including the bi-specific diabodies CD16-CD33 [54], the BiTE agent CD33-CD3 [55,56], AMG330, and the dual-targeting triple-body CD33-CD16-CD123, SPM2. Another humanized anti-CD33 monoclonal antibody, Gemtuzumab Ozogamicin, is an antibody drug-conjugate linked to calicheamicin that was validated as an AML treatment by increased clinical efficacy in a Phase II clinical trial. Patients with CD33+ AML who relapsed following treatment of gemtuzumab ozogamicin typically relapsed due to the loss of antigen, indicating a common mode of resistance [57]. Some groups have demonstrated in vitro anti-leukemic activity against CD33, but Wei-dong Han et al. were the first to report clinical use for the treatment in chemotherapy-refractory AML patients. This trial demonstrated the feasibility and efficacy of autologous CAR-T cells, which were redirected to CD33 (CART-33). Moreover, Kendrian et al. [58] also engineered CAR-T cells (CART-33) using anti-CD33 scFv that was used in gemtuzumab ozogamicin (clone My96). Results have shown significant effector functions by in vitro as well as the eradication of leukemia and prolonged survival of AML xenografts. CART33 have resulted in cytopenias and a reduction in myeloid progenitors in xenograft models of hematopoietic toxicity [58]. With encouraging results of CART33 preclinical work, Brentjens’s group took a step further to combine different elements, such as heavy and light chains of the humanized M-195 antibody, CD28-CD3ζ signaling domains, the IL-12 gene, and truncated epidermal growth factor receptor (EGFRt) [59]. With this, the transduction of the retroviral vector of tri-cistronic CAR constructs (EGFTr/HuM195-28z/IL-12) in two different preclinical mouse models has shown a reduction in peripheral CD33+ disease, indicating that further validation would be required to be implemented for clinical utilization. A novel second-generation anti-CD33 CAR that incorporated a 4-1BB-CD3ζ signaling tail has previously proven effective in clinical trials of both chronic and acute myeloid leukemia [60,61,62]. Anti-CD33 CARs have proven to be highly effective, even at very low effector to target (E:T) ratios with robust killing of both primary tumors and tumor cell lines at E:T ratios as low as <1 effector cell per 20 targets. This CAR supports a high level of T cell processivity with a single cell serially killing multiple tumor cells. As typical with early and refractory AML, this feature would be significant when the tumor burden is high. High ligand sensitivity of the anti-CD33 CAR-T cell was demonstrated with strong specific cytolysis that was independent of CD33 expression level with AML cell lines and primary AML samples. Anti-CD33 CAR-T cells were tested in an in vivo prevention model where it showed striking results with the prevention of tumor growth in established AML. In a treatment model, survival was significantly prolonged, although tumors ultimately advanced in all mice. With advanced treatment, multiple doses of CAR-T cells may be needed when an extensive disease is present to eradicate the higher tumor cell burden [58].

### 3.3. Anti-CD123 CAR Therapy for AML

The hunt for the optimal and desirable targets for AML remains enigmatic. Along with other highly expressed myeloid markers on the blast cells, CD123 was explored as a potential target. CD123 is the alpha subunit of the interleukin-3 (IL-3) receptor. Structurally, it is a single-pass type I membrane protein that contains three extracellular, one transmembrane, and one intracellular domain. CD123 is expressed on a pleiotropic range of immune cells, which include monocytes, B cells, megakaryocytes, plasmacytoid dendritic cells, and HSPCs [63]. Studies using preclinical models to validate CAR-T targeting of CD123 corroborated earlier findings using antibody-based targeting of CD123 [64], where there was no significant hematopoietic toxicity after using the preclinical model to validate CAR-T targeting of CD123 [61,65]. Scientists from City of Hope and Mardiros et al., have shown in vivo experiments with improved survival but no long-term effects which might be attributed to the retroviral vector used along with CD28 co-stimulation [65]. Gill et al. also observed that targeting CD123 via CAR- engineered T cells could result in rejection of human AML and myeloablation in immunodeficient mouse models. The CAR-T cells with costimulated 4-1BB were able to reject engrafted primary human AML in vivo but indicated impairment in the hematopoiesis in a xenograft model [66]. The primary patient-derived AML xenograft (PDX) studies which were targeted with CART123 showed almost similar effects such as tumor lysis and cytokine release syndrome (CRS) which were reported in patients with B-cell leukemia’s treated with CART19 cells [61]. Interestingly, CART123 even eliminated CD123^dim^ leukemia (UPN034) and resulted in the establishment of a T cell memory pool that was capable of rejecting disease with enhanced kinetics of expansion. In the case of CART33 or CART12, Gill et al. also experimentally observed that there was not only a significant reduction in myeloid progenitors (CD34+CD38+) but a loss of hematopoietic stem cells (CD34+CD38-) in two different humanized mouse models, also known as the humanized immune system (HIS) [67]. It has been shown that 70% of AML blasts express both CD33 and CD123, supporting the rationale for the combinatorial targeting of CD33 and CD123 in AML treatments [16]. Based on preclinical evidence, several clinical trials have been initiated to target multiple antigens [17,51,52,53]. There is strong preclinical evidence for CAR-T cells that target CD123 and CD33 antigens [54,55,56,57]. However, one major concern in the development of AML-derived CAR therapies is the potential for depletion of healthy bone marrow and progenitor cells [54,57,58,59]. These antigens were chosen because they have been repeatedly shown to exhibit elevated expression on AML blasts compared with normal hematopoietic stem cells (HSCs), indicating an essential benefit with minimal impact on normal myelopoiesis. Bispecific CARs created by the de novo production of CD33 and CD123 single-chain variable fragments (scFvs) with split CAR signals (“AND gated") exhibited optimal proliferation, cytokine production, reduced cytotoxicity, and HSC toxicity in vitro and in vivo (unpublished data Figure 2J).

### 3.4. Anti-CLL-1 CAR Therapy for AML

Expression of a novel human C-type lectin-like molecule-1 (CLL-1) was observed on the majority of AML blasts [69,70]. CLL-1 is restricted to the hematopoietic lineage, in particular to myeloid cells present in peripheral blood and bone marrow but mostly absent on uncommitted stems cells (CD34+ CD38− or CD34+ CD33−) while being present on subsets of progenitor cells (CD34+ CD38+ or CD34+ CD33+) [70]. The function of this type II transmembrane glycoprotein has been identified as inhibitory receptors, the ligands of which are yet to be known. One of the major advantages of CLL-1 is the lack of expression on other systemic/normal tissues. Jinghua Wang et al. generated a CAR using a third-generation vector system with a combination of CD28 and 4-1BB costimulatory domains against CLL-1 (CLL-1 CART cells) [71]. Preclinical observation targeting CLL-1CART cells in primary AML patient samples indicated elimination of CLL-1-expressing cells inoculated in a xenografted mouse model. The co-culturing CLL-1 CART cells generated from normal healthy donors with normal autologous HSCs enriched from bone marrow cells or PBMCs led to the effective elimination of CLL+ progenitors cells and mature granulocytes after 24 h. Though the elimination was indicated at various degrees, it is important to note that CLL-1 CAR-T cells spared the HSCs. Unlike CD123 and CD33, which are shared between AML blast and normal HSCs, CLL-1 is not expressed on CD34+ CD38− HSCs and can be potentially used as a target antigen. Wang et al. thus demonstrated that the CLL-1 CAR-T cells are a safe therapy with high potential for AML treatment. Along similar lines, Tashiro et al. described the development and evaluation of CLL-1-specific chimeric antigen receptor T cells (CLL-1.CAR-Ts) and further demonstrated their specific activity against CLL-1+ AML cell lines as well as primary AML patient samples in vitro. Compared to control T cells, CCL-1.CAR-Ts selectively reduced leukemic colony formation in primary AML patient peripheral blood mononuclear cells. In a human xenograft mouse model, CLL-1.CAR-Ts mediated anti-leukemic activity against disseminated AML and significantly extended survival. To make it more effective with a systemic approach, Tashiro et al. introduced the caspase-9 suicidal gene system to gain control over the CLL1.CART targets as well as to reduce the unwanted killing of the mature normal myeloid cells [72]. A major drawback of targeting CLL-1 is that this antigen is also variably expressed in mature myeloid cells. However, normal progenitor cells are not targeted by CLL-1.CAR-Ts, and the decline or active elimination of this effector population after therapy should enable mature myeloid cell regeneration.

### 3.5. Anti-h8F4-CAR-T Therapy for AML

Along with CD123 and CD33, there is an extensive search for other candidates that could be used for CAR-T cell immunotherapy to target AML. In AML patients, the PR1 peptide, derived from the leukemia-associated antigens proteinase 3 and neutrophil elastase, is overexpressed on HLA-A2. In previous studies, Molldrem et al. utilized PR1-induced peptide-specific cytotoxic T lymphocytes (CTLs) in normal HLA-A2+ individuals. These CTLs showed HLA-restricted cytotoxicity toward myeloid leukemias [73]. Furthermore, PR1-specific cytotoxic T cells were identified in the peripheral blood of myeloid leukemia patients, mediating cytotoxicity against leukemic blasts in vitro. In an AML xenograft mouse model, researchers reported a reduction in leukemic burden [74]. PR-1 specific CTLs were also associated with a graft-versus-leukemia (GVL) [73] effect after allogeneic stem cell transplantation (allo-SCT) [74,75]. Sergeeva et al. reported the development of a TCR-like murine IgG2a mAb, m84, which binds to PR1/HLA-A2+ AML, mediating the lysis of AML in vivo, which successfully depleted AML in vitro [76,77]. Some studies have shown that a humanized version of mouse 8F4 human IgG1 8F4–h8F4 retains antitumor activity against AML with specificity towards PR1/HLA-A2 [77]. The use of umbilical cord blood (CB) lymphocytes has been an ideal choice, as they are naïve T cells [78], since CB transplantation (CBT) was performed for malignant and non-malignant diseases [79,80]. Previously, it was reported that placental blood could be used as a source of hematopoietic stem cells for transplantation into unrelated recipients [81,82]. The transduction of T cells derived from UCB with the h8F4-CAR demonstrated their capacity for killing leukemic cells in a PR1/HAL-A2-dependent manner. Successful development and preclinical results of h8F4-CAR have the potential for a novel T cell therapy directed against an endogenous self-antigen that is differentially expressed on the cell surface of leukemic stem cells.

### 3.6. Anti-CD70-Directed CAR-T Cells for AML

CD70 is a promising target for CAR-T cell therapy against AML because it is expressed on AML bulk cells, leukemic stem cells (LSCs), and has little to almost no expression on normal hematopoietic cells (HSCs) and normal bone marrow cells [83]. CD70 belongs to the TNF-alpha family and recently has gained importance as a cell surface target antigen for AML [84]. Reither et al. demonstrated that AML patients treated with a hypomethylating agent (HMA) had upregulated CD70 expression in LSCs. With the help of a human monoclonal antibody (mAb) for CD70, cusatuzumab, an attempt was made to block CD70/CD27 interaction. This study has shown promising in vitro and in vivo (PDX model) results by eliminating LSCs via antibody-dependent cell cytotoxicity (ADCC). These preclinical results led to a phase I clinical trial in combination with HMA azacytidine. Disappointingly, phase II resulted in cusatuzumab CR rates being less than half of those seen in phase I [85]. Recently, Sauer et al. have shown the potential of CD70-specific CAR-T cells against AML without causing HSC toxicity [86]. In this study, a comparative analysis for CD70 CAR (full length-CD27 fused to CD3ζ) was done among the CD70scFv CAR-T cells. Among the parameters studied to evaluate these CARs, enhanced tumor activity, as well as higher proliferation potential, was indicated by CD27z-CAR-T cells. A study led by Qong J Wang et al. utilized second-generation CARs that included the 4-1BB costimulatory domain, which was thought to be superior to first-generation CARs [87]. Among the various constructs used within this study, 41BB and CD3ζ (trCD27-4 1BBζ) CAR-T cell constructs conferred the highest IFNγ production against CD70-expressing tumors in vitro. Similarly, an in vivo xenografted mouse model with human tumors expressing CD70 antigen showed promising results. At the American Society of Hematology (ASH2021) meeting, an ongoing study in Marcela Maus lab on CD70-directed CAR-T cells for AML was presented [88]. The investigation has used the NSG Molm13 AML model to assess CAR-T cell anti-tumor activity. CD70-CAR-T cell activity was attenuated due to the soluble form of the ligand, CD27, that was confirmed in AML co-culture assay. To overcome this limitation of the soluble ligand, a range of novel hinge CAR variants were evaluated (truncated, deleted, flexible, and “CD8hinge&TM”). Compared to 41BB-based native CAR-T cells, the CD8hinge&TM variant has shown marked improvement in the in vitro cultures with higher cytolysis of AML targets. Moreover, the CD8hinge&TM CAR-T cells had shown superior in vivo expansion and were able to mediate elimination AML in the PDX model compared to the native CAR-T cells.

### 3.7. Anti-TIM-3 CAR Therapy for AML

Even after chemotherapy, the reason for the relapse of leukemia has been linked to the rare chemo-resistant fraction of LSCs that proliferate to develop leukemia blasts. So far, none of the conventional therapies or treatment strategies have been successful because they lack targeting and eliminating LSCs, thus enabling the residual clones to gain the capacity to outgrow the treatment. Recently, the role of T cell immunoglobulin mucin-3 (TIM-3) was discovered in T cell exhaustion, which also correlated with the outcome of anti-PD-1 therapy [89]. Along with CTLA-4, TIM-3 is another immune checkpoint that could inhibit cancer immunity. TIM-3 as a surface molecule is expressed on LSCs in almost all types of AML but not on HSCs. Yoshikane Kikushige et al. have shown that, when immunodeficient mice were injected with TIM-3(+) but not TIM-3 (−), AML cells led to the reconstitution of human AML [90]. This indicated that the TIM-3(+) population contains functional LSCs that could give rise to the disease and blasts. Targeting TIM-3 to treat MRD in patients with AML would provide a clinical benefit. TIM-3 is highly expressed on AML blasts and LSCs in most subtypes of the patients. The lack of expression in normal tissue as well as the granulocytes, lymphocytes, and most HSCs make this protein an ideal target. Wen-Hsin Sandy Lee from the Dr. Wang group has recently generated a phage library using naïve human Fab and was able to isolate anti-human TIM antibodies. Utilizing scFv from this antibody, a second-generation CAR-T cell redirecting TIM3 was developed, which displayed antileukemia activity against cell lines as well as primary AML blasts [91]. Anti-TIM3 CAR-T cells were able to suppress leukemic cell growth in vivo. The NSG was engrafted with the CMK cell line, thus generating a xenograft model. Bioimaging measuring luciferase activity indicated infiltration of the cell line in bone marrow, spleen, and liver. During the treatment, suppression of tumor growth was observed, which suggested tumor clearance. Levels of the cytotoxicity effectors such as IFN-γ, granzyme B, perforin, GM-CSF, and IL-5 were significantly elevated compared to the control group (A11 developed tumor model). The study demonstrated efficient killing of the primary LSCs directly isolated from the patients. Thus, anti-TIM-3 CAR T cell therapy following the first-line treatment may improve the clinical outcomes of patients with AML. Xin He et al. had also presented their work on bispecific and split CAR-T cells, TIM3, and CD13 to target and eliminate AML [92]. In addition, the sequentially tumor-selected antibody and antigen retrieval (STAR) method could identify nanobodies that would bind AML, STAR-isolated nanobodies Nb157 are specifically bound to CD13, which is highly expressed in AML cells, and CD13 CAR-T cells potently eliminated AML in vitro and in vivo. CAR-T cells bispecifically targeting CD13 and TIM3, which are upregulated in LSCs, eliminated patient-derived AML while reducing the cytotoxicity of stem cells and peripheral myeloid cells in mouse models. These findings could lead to the development of an effective AML CAR treatment.

### 3.8. CD93 as Target for CAR-T AML

CD93 is a C-type lectin transmembrane receptor that plays a role not only in cell–cell adhesion processes but also in host defense. Coustan-Smith et al. have published a study that elaborately explains the monitoring of minimal residual disease (MRD) in AML [93]. The report considered a genome-wide expression of AML cells from patients and compared it with healthy individuals. The study has outlined two major data sets, which are based on the aberrant expression of genes followed by flow cytometric analysis of leukemic and non-leukemic myeloblasts from bone marrow. Nearly twenty-two putative markers have shown aberrant expression on AML samples. Many of these markers are well known in leukemogenesis (leukemia-associated markers) and are currently used for the identification of AML (leukemia diagnosis) [94]. Among these 22 markers, CD93 has an interesting expression profile. Though CD93 is not expressed on HSC or any other precursor myeloid population, it is apparently expressed on other range of immune cells, including neutrophiles (CD15), monocytes (CD14) and mature myeloid cells [95]. Lack of expression on B cells (CD19), T cells (CD3), RBCs (CD235a) and platelets (CD41a) has made the CD93 marker of additional interest. Recently, Richards et al., in collaboration with Dr. Majeti, designed CAR-T cells, which have shown anti-leukemic strength, targeting CD93 using a novel humanized CD93-specific binder [96]. The in vivo model revealed no toxicity towards HSC and progenitor cells. Although there is a lack of expression of CD93 on HSC, it is widely expressed on tissue resident human endothelial cells, and CD93 CAR T cells can identify and elicit an “off-target” response to endothelial cell lines. To address this challenge, another Boolean strategy was applied to avoid endothelial-specific cross-reactivity. The authors have provided proof of concept for “NOT-gated” CD93 CAR-T cells that circumvent endothelial cell toxicity in a relevant model system. Briefly, CD93 CAR-T cells can specifically eliminate AML and spare HSPCs but exert on-target, off-tumor toxicity to endothelial cells. Uniform Manifold Approximation and Projection (UMAP) clustering indicated the expression pattern of CD93 on human enodthelial cells, which was supported by IHC evaluation of tissues from various organs, such as the pancreas, skin, diaphragm, and heart. To avoid the targeting of enothelial tissues and causing toxicity, the authors preferentially designed CARs with NOT-logic gate. An in vitro model was generated to test the specificity of this inhibitory CAR-based NOT CAR construct (iCAR). This model is based on the immortalized human umbilical vein endothelial cell (iHUVEC) cell line expressing truncated CD19 (extracellular and transmembrane domain): iHUVEC-19, which is targeted by CD19-specific iCAR. The design of the novel iCAR included CD19-specific scFv, an CD8alpha transmembrane signaling endodomain from immunoreceptor tyrosine-based inhibitory (ITIM)-containing proteins, including PD-1 and T cell immunoreceptor with Ig (TGIT). No intracellular signaling domain (Pdel) was included as the control. T cells expressing activation CARs (aCAR expressing endodomain CD28 costimulatory domain with CD3ζ targeting CD93) co-expressing iCARs were cocultured with THP-1, iHUVEC and iHUVEC-19, respectively. Comparative analyses between iCAR/aCAR and control constructs (CD93-28-CD3ζ/Pdel-1) demonstrated significantly less IFNg and IL2 secretion when cocultured with iHUVEC-19, whereas similar levels of these cytokine secretions were reported when cocultured with the THP-1 (AML) cell line. TGIT-based iCARs have been shown to induce the inhibition of cytokine in the presence of two targets, CD19 and CD93, whereas the construct retains the functionality after the identification of both iHUVEC and iHUVEC-19 cells, indicating baseline antigen-independent inhibitory signaling. The authors have identified another 232 candidate targets for an iCAR-based NOT-gate work. In-depth validation of iCAR would be required based on cell surface expression on normal cells. It is imperative to confirm lack of expression target antigen and modulation of the design to target AML.

NOT-gate target antigens selectively expressed on the cross-reactive tissue to propagate an inhibitory signal that interferes with the CAR T cell activation signal [97].

Overall, the study has shown co-expression of other AML targets on endothelial cells, introduced a novel NOT-gated strategy to mitigate endothelial toxicity, and demonstrated the use of high-dimensional transcriptomic profiling for the rational design of combinatorial immunotherapies. While the proof-of-concept study has demonstrated the significance of NOT-gate-based CARs, a PDX model would be required to translate the understanding and resolve the new set of challenges.

## 4. Limitations and Challenges

### 4.1. Toxicities

CAR-modified T-cells are potent and have demonstrated potential for “on-target off-tumor” toxicity in cases where the antigenic target is co-expressed on normal cells such as with leukemia-associated antigens. Major clinical on/off tissue toxicity effects for CD19-targeted CAR-T cells include cytokine release syndrome (CRS) [98], which is caused by activation of T cells and manifests as high-grade fever, hypertension, and immune effector cell-associate neurotoxicity syndrome (ICANS) which is associated with endothelial cell dysfunction at the blood-brain barrier caused due to hyperinflammation [99,100]. Additional effects include, cardiovascular complications, capillary leak syndrome associated with hypoxia, occasional renal dysfunction and coagulopathy have been reported [2,101,102]. In a recent review about the CRS and ICANS in cancer immunotherapy, Morris et al. have highlighted the pathophysiological causes behind the neurotoxicities and development of novel therapeutics for the prevention and/or management of these toxicities [98].

### 4.2. Loss of Antigen

Although there has been rapid progress of CAR-T cell therapy, resistance to treatment has become a major barrier for its clinical application. Even though the key Novartis CAR trial with CTL109 (NCT01626495) has shown a remarkable response rate of 93% [103,104], the 1-year complete response rate has decreased to around 55% and suggested that almost half of the patients developed resistance in that period. Target antigen loss accounted for most of the resistance in CAR-T cell therapy. This demonstrates that targeting a single antigen may not be sufficient for any CAR-T cell therapy. Relapse from epitope-loss variants or lineage switch after CD19 CAR-T cell therapy against ALL has already been reported [105,106] and few have shown tumor-antigen editing in response to immunotherapy [107]. Most of the CAR-T cell clinical benefits have come with patients suffering from recurrent ALL, especially in the case of Emily Whitehead, who currently has nearly 10 years of cancer-free survival. Potential advantages of CAR-modified T cells over monoclonal antibodies include greater cytotoxicity, active trafficking, passage through the blood-brain barrier, fewer required doses, the potential for long-lived memory and protection against relapse, and increased sensitivity to low antigen density [63,101,108].

### 4.3. Preclinical Models

Although in vitro basic functions such as proliferation, antigen target lysis, cytokine production, and antigen-based stimulation assay are quantified, these assays do not necessarily turn out to be efficiently translated with in vivo performances. The validation of CAR-T cell design was done through engrafting the preclinical mouse model with adoptively transferred T cells. Though these mouse models have shown an improvised efficiency for clearing tumor cell burden, the immune microenvironment inside a mouse and human lymphoid organs largely differ from each other. Due to these limitations, an alternative method for evaluating CAR-T cells that can be adoptively transferred into immunocompetent mice was used. This mouse model does not necessarily translate well to human T cell performance. Initially, the efficiency of CAR-T cell therapy in mouse models severely lacked to translate into clinical trials [49,109,110]. Lastly, mouse models also have a drawback where they cannot assess the cross-reactivity against the healthy human tissues that might express the target antigen systemically at low expression levels.

## 5. Future of AML CAR-T Cell Therapy

Currently, there are approximately more than 190 ongoing clinical trials in the field of immunotherapy. The functional assessment of CAR-T cells has been progressing forward, where the results from these experiments have been formulated into phase I clinical trials. With every trial, significant progress has been made in the understanding of CAR-T cell biology. Current efforts are being made in designing construct as well as other areas to build a robust model system with superior anti-leukemia activity with a lower cell cytotoxicity (Figure 3).

### 5.1. Fine Tunning

Structural and physiological challenges through previous work have warranted these modifications of CARs. Multiple approaches have been taken to modify the design to enhance specificities for tumor targeting, such as fine-tuning CAR-T cells synergistically, providing additional immunomodulatory proteins to enhance potency, improvising cell trafficking, and persistence to advance CAR-T cell therapy. Detailed reviews that summarize all these factors are readily available in the literature [111,112]. In short, modification of the structural backbone can be done via an additional co-stimulator, inhibitor or a few regulatory domains to enhance the function. The functional fine-tuning of CAR is done through bioengineering approaches to enhance and mitigate CAR-mediated toxicity. Specific efforts are being made concerning spacer length, transmembrane domains, signaling sequence and targeting multiple antigens through a variety of “gating” strategies. As previously mentioned, some studies involved an additional transgenic inducible secretion of the inflammatory cytokines that has been included in the construct, which is meant to modify the tumor microenvironment (TME) by enhancing antigen cross-presentation and promoting epitope spreading. These T cells are redirected for universal cytokine-mediated killing TRUCK or “armored” CAR-T cells by expressing IL-12 [59,113], IL-18 [114], or CD40L [115,116].

### 5.2. Enhancing Safety and Minimizing Toxicity

The idea of a multi-antigen-targeting strategy is based on Boolean logic. The Boolean logic is applied to “gate” the activity of CAR-T cells to maximize the potential of binding the target antigen and reduce the off-tumor toxicity to the bystander cells. The logic gates are an assembly of two different surface receptors expressed on a CAR-T cell to identify two separate antigens to discriminate between target cells and healthy cells. One strategy incorporates two separate receptors, one with an endodomain of CD3ζ, whereas the other receptor contains a co-stimulatory domain. Kloss et al. have shown dual chimeric receptor-mediated activation and co-stimulation of human T cells that led to robust cytotoxicity and efficient reduction in tumor burden targeting PSMA and PSCA [117]. This study led to the excavation of the concept of the “tumor sensing” strategy. This strategy could help avoid the off-tumor side effects, as the tumor with both antigens will be selectively eliminated compared to the single antigen. The (i) “OR” logic gate is where the completely independent CAR molecules are installed to recognize the presence of a single antigen or both antigens simultaneously. Contrastingly, multiple targeting-complete CARs (with co-stimulatory domains) are pooled to co-express specifically to identify target antigens, which enables CAR-T cells to prevent tumor escape by targeting either of the antigens. (ii) “AND” logic-gate: two distinct CARs are co-expressed on a single T cell with complementary signaling domains; of the two CARs, if one meets the target with the absence of the other target, the CAR-T would be rendered inactive. The CAR-T cell would only obtain full activation status upon binding their cognate antigens simultaneously. Roybal et al. provided a proof of concept with the meticulous designing of combinatorial antigen recognition T cell circuits, synthetic Notch (synNotch) receptors [118] (Figure 2B). With the recognition of the first antigen by the first receptor, a transcription factor would trigger the induction/expression of a second CAR, which will then bind to the secondary antigen, thus completing the circuit. In short, these dual-receptor CAR-T cells are only armed and activated after binding to both antigens on cancer cells (AND-gate) [119]. Intricate experimental designs targeting two antigens with synNotch (dual CAR-T) could bind synergistically and selectively, to avoid on-target off-tumor toxicity activate the CARs. The in vitro experiment with the bioengineering of the dual specific targeting of CD19 and mesothelin has shown that synNotch CAR strategies would elicit an antitumor effect [120]. While these multi-CAR systems are an exciting and important approach to enhance tumor targeting, a wider range of combinatorial-sensing strategies will improve the ability to treat a variety of tumors and diseases with T cell therapies [121]. (iii) Furthermore, through the implementation of the “NOT” logic gate (Section 3.8), the T cells can be engineered to differentiate target cells from non-target cells, which will avoid attacks on normal tissue and result in the achievement of the safety of the CAR-T cells.

A Universal CAR-T (UniCAR) is derived from second-generation CAR-T cells, which utilize CD28 as a costimulatory domain (Figure 2F). The design is divided into two components. First, a unique scFv that does not directly bind to the cancer cell (TAA) but identifies a peptide motif on an adaptor molecule. Second, this soluble adaptor molecule called a “targeting module” (TM) is configured to identify and bind CD123, which ligates at one end to the TAA and on the other end the CAR-T receptor. The TM confers specificity against TAA and due to its high flexibility for TAA binding, it can target heterogeneous hematologic malignancies such as AML. A preclinical study with the UniCAR-T system has shown that even small amounts of TM can induce the lysis of tumor cells along with the release of the cytokine [88]. This study led by Cartellieri et al. provided a “proof-of-concept” for UniCAR retargeting human UniCAR-engineered T cells against the AML antigens CD33 and CD123 using a novel modular platform. The UniCAR02-T-CD123 drug is a combination of a cellular component (UniCAR02-T) with a recombinant antibody derivative (TM123), which together form the active drug. A phase I clinical trial using UniCAR T-cells targeting CD123 for the treatment of CD123-positive hematologic and lymphoid malignancies has delivered some impressive results (NCT04230265) [89].

A split, universal, and programmable (SUPRA) CAR system for T cell therapy has the ability to switch targets without re-engineering the T cells, finely tune T cell activation strength, and sense and logically respond to multiple antigens [120,122] (Figure 2E)). Due to the fine-tuned T cell activation strength, a reduced cytotoxicity was observed while performing multiple functions. SUPRA CAR can sense and logically respond to multiple antigens to combat relapse as well as the ability to control cell-type-specific signaling. The highest advantage of SUPRA CARs is in its ability to impart minimal immunogenicity while targeting tumor-associated antigens. Programmable controllability in the design of the construct, is due to two-component system, thus termed as split-CARs. The split-CAR system is composed of “zipCAR” and “zipFv” fragments (Figure 2E). The zipCAR contains intracellular signaling domains connected via a transmembrane segment to an extracellular leucine zipper. The zipFv contains a ligand-binding scFv domain fused to a second leucine zipper. A functional CAR is reconstituted when zipFv proteins are added to engineered T cells that express zipCARs with matching leucine zippers.

### 5.3. Modulation of the Tumor Microenvironment

The inclusion of “accessories” along with CAR, such as immunomodulatory factors leading to the secretion of cytokines, chemokine receptors, synthetic receptors, and the blockade of checkpoint inhibitors (anti-PD-1 scFv), has enhanced the CAR-T cell efficacy. Reviews of the choice of co-stimulatory domain and the differences in various activity, persistence, metabolism, memory phenotype and extensive characterization, as well as immunophenotyping, have been well documented for CARs [61,123,124]. The new list of co-stimulatory domains and their functional repertoire are actively being investigated [125]. Some of these consist of CD40, OX-40 [126] (CD134), ICOS (CD278) and CD27 [127]. Lastly, focused research is done on reversing the exhausted state of CAR-T cells as well as persistence/maintenance of CAR-T cells as a memory subset to harvest better therapeutic efficacy. T cell exhaustion is characterized by an increased expression of inhibitory receptors, as well as widespread transcriptional and epigenetic alterations [128]. Eyquem et al. showed that inserting CAR transgenes using CRISPR/Cas9 genome editing technology into the T cell receptor α constant (TRAC) locus ensued in CAR expression in T cells [129] (Figure 1: fifth generation of CAR design). This improvement has resulted in enhanced T cell potency, compared to conventional CAR-T cells in a mouse model of ALL.

### 5.4. Re-Energizing CAR T Function

Re-expression of the CAR, followed by single or repeated exposure to antigen, delayed the effector T cell differentiation as well as the pattern of exhaustion [130]. Overexpression of the transcription factor c-Jun has been shown to protect T cells from exhaustion from even the most exhausting CAR designs [131,132]. Weber et al. have shown that chronically exhausted CAR-T cell expanded clones through tonic signaling have distinct hallmarks of exhaustion [133]. An inert small molecule dasatinib, tyrosine kinase inhibitor, which suppresses T cell activation via the inhibition of proximal T cell receptor (TCR) signaling kinases, such as Src, Fyn, and Lck, was approved by the FDA [133]. In pre-clinical and clinical studies, dasatinib has been shown to prevent or reverse T cell exhaustion [134]. A detailed study has outlined that the state of transient inhibition of CAR-T cellular signaling in the presence of dasatinib revealed a memory-like phenotype with enhanced anti-tumor activity in an adoptive transfer into a xenograft-mouse model. Pre-exhausted CAR-T cells restored anti-tumor functional phenotype with the induction of “rest” for as few as 4 days, indicated through transcriptional reprogramming and epigenetic remodeling. The duration of “rest” was associated with decreased expression of the exhaustion-associated transcription factor TOX and elevated expression of memory-associated transcription factors LEF1 and TCF1. Functional rejuvenation was dependent on the activity of the histone methyltransferase EZH2, which was consistent with epigenetic remodeling in response to “rest” (Figure 3A). In conclusion, a small period of “resting” or inhibition of CAR signaling could restore the activity of CAR-T cells as well as enhance CAR-T cell fitness by preventing exhaustion. The regulation of modified CAR-T cells with intermittent inhibition could translate in clinical CAR-T cells. A recent review by Weber et al. has covered most of the upcoming strategies associated with immune cell therapies [135]. Lui et al. introduced a genetically engineered switch receptor construct, comprising the truncated extracellular domain of PD1, the transmembrane, and cytoplasmic signaling domains of CD28 into CAR-T cells. This “switch” receptor converts suppressive signals induced by tumor PD-L1 into activation signals [136].

### 5.5. Extensible and Adaptable CARs

These promising results have demonstrated the functionality and switchability of adaptor CAR T cell systems for the first time in humans in a clinical trial [137]. Given the heterogeneity of antigen expression on AML cells, combinatorial targeting approaches are required to improve therapeutic tools. Using modular Ab-based systems, a switchable, flexible and programmable adaptor Reverse CAR (RevCAR) platform was designed [138] (Figure 2G). The UniCAR system led to the foundation of this study, where tumor specificity was controlled with a reversible affinity of adopter molecule giving a joystick to control safety in the CAR-T therapy [116,139]. In this study, Enrico Kittle-Boselli et al. developed a RevCAR T cell that can indirectly target tumor antigens, which can be controlled by the binding of bispecific antibody cross-linking RevCAR-T eventually resulting in tumor lysis. In short, RevCARs express peptide epitopes E5B9 or E7B6 instead of a scFv conventional second-generation CAR. These RevCARs are unable to directly recognize tumor antigen, instead they cross-link between bispecific target modules (RevTMs). These RevTMs consist of two scFvs: one which targets the respective RevCAR E5B9 or E7B6 epitope and the other that targets tumor antigen. The adaptor modular in this system can act as a control that allows for an on/off switch of the RevCAR T cell activity by dosing the RevTMs. The reversibility of this system not only provides an advantage of specific targeting but adds value while avoiding on-target off-tumor toxicities such as CRS.

## 6. Conclusions

In the last few years, CAR-T cell therapies have emerged as a new class of investigational 697 therapy, showing impressive results in hematological malignancies, especially in B cell malignancies. In the late 1980s, Eshhar et al. conducted experiments that served as the foundational pillars of immunotherapy and genetic engineering. The collective efforts by various groups worldwide, through preclinical mouse models, have resulted in significant improvements in the treatment of CAR-T cell therapy. With a better understanding of the biology of CAR-T cells, some multicenter clinical trials of these improved/modified immune cells, termed CAR-T cells or “living” drugs, were used to treat patients with hematological malignancies. This success paved the path for FDA approval of CAR-T cell therapy for the treatment of B cell malignancies. So far, the success has been limited only to lymphoma, and treatments for leukemia remain at large. Learning through each trial and clinical management study has provided new challenges for these therapeutic agents. In current landscape of CAR-T cells technology, efforts are carried out to synergize various platform like radio-immunotherapy [140,141]. New acquired list of targets for solid tumor and various CAR modifications have been described elsewhere [142,143] We have listed above selective promising targets, and efforts that have been made in preclinical models. The success of bioengineering approaches, including Boolean gating, reversing exhaustions, ectopic overexpression of transcription factors, multi-specific “adaptors”, SUPRA CAR, switch CARs, and synthetic gating platforms, will ultimately determine the extent to which next-generation immune cell therapies emerge as efficacious alternatives to traditional medicines.

## Figures and Tables

**Figure 1 cancers-14-01241-f001:**
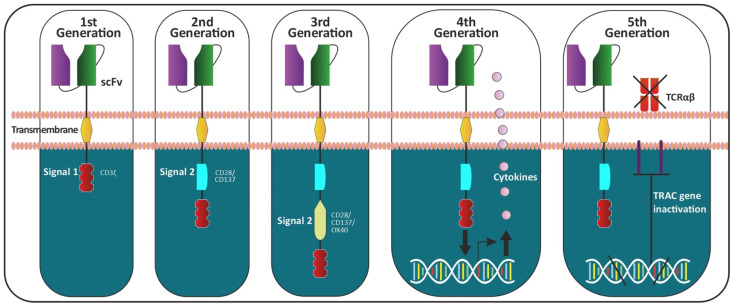
Generation of CAR-T cells. The CAR-T cells are composed of three components: (i) extracellular domain: antigen-recognition domain is a single-chain fragment variant (scFV) fusion protein of the variable regions of the heavy (V_H_) and light chains (V_L_) of immunoglobulins, an antigen-specific immunoglobulin separated by a flexible linker, (ii) the transmembrane domain and (iii) the intracellular domain (endodomain: containing several functional units). Initial efforts were made to include only one intracellular signal component CD3ζ (which contains three immunoreceptor tyrosine-based activation motifs (ITAMs), which is important for signal transduction) in the first generation of CARs. The second generation was composed with an addition of a single costimulatory molecule (CD28 or 4-1BB) to the first generation. Various groups co-founded this preclinical research to determine the efficacy projected by including co-stimulatory receptors. The combining of two separate co-stimulatory domains to the second generation of CARs led to the evolution of the third generation of CARs. The fourth generation of CAR-T cells can activate the downstream transcription factor to induce cytokine production (constitutively or inducible expressing inflammatory cytokine: IL-12 or IL-18). These T cells are also referred to as T cell redirected for universal cytokine-mediated killing (TRUCKs). The fifth generation of CARs, based on the second generation, uses gene editing for the insertion of CAR (CRISPR/Cas9) into the TRAC gene for inactivation, leading to the removal of the TCR alpha and beta chains.

**Figure 2 cancers-14-01241-f002:**
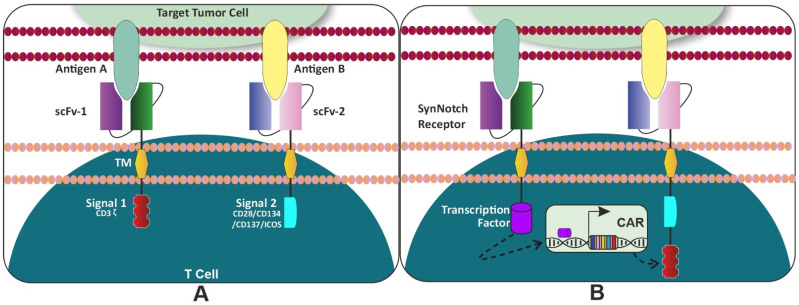
Modifications of CAR-T cells: (**A**) Logic gate strategies: One CAR molecule for activation and a second CAR molecule for co-stimulation in the same T cell: combinatorial antigen recognition is optimized to reduce the “on-target/off-tumor” toxicity. (**B**) SynNotch “AND” gate circuit: First the synthetic Notch receptor will identify target antigen A, then the terminal cytoplasmic domain on the proximal side of synNotch-CAR “transcription factor” will get cleaved and enter the nucleus and finally induce the expression of CAR. The second CAR receptor will bind to antigen B to complete the circuit. (**C**) “ON” Switch: In the presence of a small molecule, two parts of the CAR can be reassembled, at the split construct, thus activating the CAR-T cells. The design strategy of an “AND” logic gate requires both “antigen + small molecule” for T cell activation. (**D**) “OFF” Switch: Armed CAR-T cells with “OFF switch”: suicide gene (iCasp9) is required to sequentially activate the kill switch by the mechanism of apoptosis. iCasp9 is co-expressed with the CAR molecule in CAR-T cells. Addition of a small molecule, AP20187 (CID: chemical inducer), leads to dimerization of iCasp3, which triggers downstream Casp3 to induce apoptosis. (**E**) SUPRA CAR: A zipFv has an scFv linked to a leucine zipper (Zip-A). A zipCAR has a cognate leucine zipper (Zip-B) that can bind to the Zip-A. The binding of A and B leads to activation of the T cells. SUPRA-CAR acts as a universal CAR. The affinity between the A and B-leucine zippers is tuned so that the signaling strength and activity can be controlled. The binding of Zip-A to a null antigen will abrogates the signaling cascade, and thus rendering T cells to an inactive state. (**F**) UniCAR CAR: Universal CAR-T (UniCAR) platform that can be rapidly turned on and off with a switch consisting of two components. First, CAR does not directly recognize the tumor antigens and thus is inactive. The second component is a targeting module (TM), a soluble adaptor that comprises a motif recognized by the UniCAR’s receptor, and a highly flexible antigen-binding moiety directed against a cancer antigen. (**G**) RevCAR platform: Composed of extracellular peptide epitope E5B9 or E7B6 and CD28, hinge domain (HiD), CD28 transmembrane domain (TMD), intracellular CD28 costimulatory domain (CSD) and CD3ζ activating signaling domain (ASD). RevCAR-E5B9-28/3z or RevCAR-E7B6-28/3z CAR-T cells are redirected towards CD33 or CD123 expressed on AML blasts via adaptor target modules, named “RevTMs”. RevTMs are bispecific antibodies (bsAbs) consisting of two different single-chain variable fragments (scFvs) binding on the one arm to E5B9 or E7B6 of the RevCAR and on the other arm to CD33 or CD123 on the surface of AML blasts. (**H**) STOP-CAR platform: The S chain is composed of a c-Myc and DAP10, a transmembrane domain, a costimulatory domain, Bcl-XL, and CD3ζ. The R chain is comprised of an scFv, a transmembrane domain, a costimulatory domain, and human apolipoprotein E4 (apo4). When the disruptive drug is not administered, the R and S chain can bind to each other, and the CAR can be activated upon target antigen engagement. After the disruptive drug administration, it secures itself to its binding site on the Bcl-XL domain located on the S chain. Due to this, it renders the R and S chain’s ability to pair and the CAR becomes un-activatable. (**I**) Proof of concept for NOT-gated CD93 CAR T cells that can mitigate endothelial cell toxicity (**J**) Novel bispecific human CD33/123 CAR T cells with split signaling “AND gate” costimulation domains were highly efficacious in vitro at killing target cells, proliferating and generating substantial amount of cytokines (unpublished data) [68].

**Figure 3 cancers-14-01241-f003:**
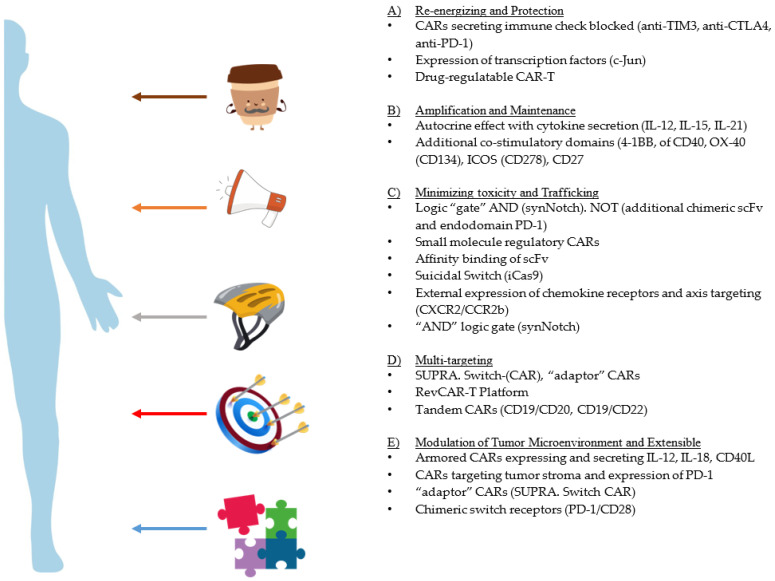
Preclinical Bioengineering Strategies to Improvise the CARs: An experimental approach with immunocompromised in vivo models is underway to enhance specificity and cognate antigen targeting. These efforts are mitigated to reduce toxicity and increase the validation of the preclinical model. Few of these modifications are under investigation to reverse the exhaustion. (**A**) Re-energizing of the CAR-T cells has led to the increased persistence as well as reverse resistance to secondary exposure of antigens. Chronic activation of CAR-T cells leads to exhaustive phenotypes. Transient inhibition (“rest”) of CAR signaling has restored the functional phenotype, which was demonstrated by the transcriptional reprogramming and epigenetic remodeling. Enforced ectopic expression of c-Jun and deletion of transcription factor (NR4a) leading to exhaustion have reversed and restored the anti-tumor to an extent. Armored CARs secreting immune checkpoint blocks will be further enhanced by unleashing the breaks on the CAR-T cells. The most important modification identified in all clinical trials consisted of the CAR-mediated cytotoxicity. (**B**) The combination of CAR secreting IL15 and oncolytic viruses have been recently configured to eliminate tumor cells. Evolution of the third-generation CARs with the integration of multiple co-stimulatory domains (e.g., CD137 (4-1BB)) exhibited better persistence of CARs compared to CD28. The fourth-generation CARs have the ability to constitutively secrete cytokines, such as IL-12, IL-18, IL-21, which have the ability to amplify and expand the CARs as well as autoregulate in a specific environment. (**C**) To reduce this cytotoxicity, suicide switches (iCasp9), adaptor CARs (RevCARs), and Switch-CARs were engineered. A small molecule and regulatable CARs with an on/off switch were developed to minimize further toxicity. “AND” logic gating, (e.g., synNotch) and bispecific CARs were developed to reduce “on-target off tumor cytotoxicity”. Overexpression of CXCR2/CCR2b axis and CXCR5 were expressed to redirect CAR trafficking to the tumor site and increase mobility to systemic locations. Co-transduction of CARs with various scFvs targeting multiple antigens was designed to reduce cytotoxicity and increase tumor binding specificity. (**D**) Tandem CAR scFvs (e.g., CD19/CD20, CD19/CD22) were utilized to improve target specificity. Armored CARs secreting IL-12 ad IL-18, as well as CARs targeting the tumor stroma, have recently made an immense difference in the modulation of the tumor microenvironment. (**E**) Split-CAR design that enables the engineering of multi-feature CAR-T cells, aiming to address current challenges limiting the safety and efficacy of CAR-T cells for cancer treatment (a split, universal, and programmable (SUPRA) CAR).

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
