# Peer review of "Emerging CAR T Cell Strategies for the Treatment of AML"

_cancers, 2022, doi:10.3390/cancers14051241_

Round 1

Reviewer 1 Report

The authors have done a comprehensive search of the literature and this is a really well written review on a very interesting subject that hopefully will evolve to a novel clinical concept.

I have only a few minor points:

-In the abstract the authors mention that CAR-T therapy can eliminate AML. While there is strong preclinical evidence that this may be possible, at this point clinically this approach has not yet been particularly successful so I would not say that, instead can say that this is a promising novel approach for AML.

-The authors have forgotten to add refs in a few points where it still reads as (ref) - Lines 131, 317, 658

Author Response

thank you for your encouragement and valuable comments. As you have pointed out some references were missing, we have figures those as well as revised the figure legends to provided a better understaning.

Reviewer 2 Report

This review by Vishwasrao P et al. is well written and gives a comprehensive overview of CAR-T cell therapies for AML with description of targets used for AML. The topic of the review is of high importance and this field has seen a rapid advancement in the recent past. However, for the very same reason, this has been recently reviewed comprehensively (Maucher M et al., Cancers, 2021) reducing the impact of this well-attempted review.

Thus, I am not convinced by the title of this review which suggests that the review will focus on targets for AML but it’s actually not the case. Indeed, some targets are missing so the review is not exhaustive about this. In addition, authors describe at some length the modifications of CAR-T cells and future perspectives with figures and examples. I think that the authors have to focus on this and adapt the title consequently. For me, targets selected by authors are not the main relevant interest of this manuscript.

General comments:

Lines 55-56: “ … provide a T cell with new antigen specificity.” -> “… provide a T cell with an alternative antigen specificity.” is more correct regarding that T cells keep their own TCR. Only, CRISPR CAS9 approach leading to endogenous TCR KO leads to a new antigen specificity.

Line 58: “Major Histocompatibility Complex (MHC)”

Line 60-62: “This resistance is avoided due to the functional nature of the MHC-independent to the functional nature of the MHC-independent mode of action of CAR-T cells”: correct the sentence

Line 62-63: “ T cell receptors (TCR)-ζ “ Zeta chain do not exist in the TCR. However, there is a zeta chain among the CD3. Thus, CARs incorporating CD3zeta signaling domain.

Line 67: “Administration (FDA) for the treatment of CAR-T cell therapy.” No. FDA validated CAR-T cell therapy for the treatment of multiple myeloma.

Lines 68 – 76: Authors resume the method of CAR-T therapy. However, an important step is missing. Patients are conditioned before CAR-T cells infusion in most of cases. Please add this step.

Line 613: “anitgens” please correct

Line 631 – 634: “NOT” need to be more explain and add clearly in the related figure.

References:

Authors must put the reference at the end of each scientific fact that they use. Here some sentences where ref are missing. It is not complete; authors must correct this for the entire manuscript.

Line 131: Ref is missing

Line 276 to 287: There are no references.

Line 317: Ref is missing

Line 333 to 345: There are no references.

Line 356 – 362: There are no references.

Line 404 – 418: There are no references.

Line 508; Ref is missing.

Line 533; Novartis CAR trial with CTL109, authors have to add the trial number (NCT).

Line 586: “Detailed reviews that summarize all these factors are readily available in literature.”, authors have to add references for some of these detailed reviews.

Line 658: Ref is missing

Line 666; Ref is missing

Figures:

Nothing is in place at least in my manuscript version, numbers and legends are missing. The figure 3 is cut in 3 parts (?). Authors must organize clearly figures and legends in the manuscript. The Figure numbers need to be add in the text. We do not know when we have to look at the figure when we are reading the text.

In addition, the text in the figures is too small; authors have to increase the size.

From Line 131: Figure 1: authors describe the fifth generation of CAR-T with truncated IL-2 receptor. This is nice. However, the figure describe TRAC gene activation and nothing is mentioned in the text.

Line 574 and after: Authors describe Future of AML CART-T cell therapy with an interesting figure, for me figure 2? (see previous comment regarding the disorganization of the figures). However, authors do not describe precisely all the subsets of this figure in the text. For example, SUPRA CAR and STOP-CAR platform are not explain and describe in the text as well as “on” “off” switch. Conversely, “Not” logic gate describe in the text could be add in the figure.

Last figure, name 3 in my file could be nice because it summarized CART modifications and improvements. However, the legend is not related to this figure and may be much more related to the previous one.

So, a big effort to link figures and text is required to improve the quality of this manuscript.

Author Response

Hi Reviewers,

Thank you for your esteem time and efforts to provide us with your valuable input. We have addressed most of the comments and looking forward to hearing from you. Our efforts as I have explained as well as to bring out the lasted in the field of CAR- T cells redirected towards AML. We are well aware that there is a pleiotropic list of targets and even more coming out as well write and communicating. This is our humble effort to bring out the current preclinical efforts that have been successfully found their way in journals and publications. The list is exhaustive so we have a clear mindset of providing the “Emerging CAR T cell strategies for the treatment of AML”. We not only focus on some challenging targets but have in-depth provided ways in which efforts are put together by many labs to putatively target more than one tumor antigen to increase the specificity of the CAR-T cells.

We would like to know which other targets the would reviewer like us to describe and contribute further.

  • We would take all suggestions into consideration. We found a few of these targets more amicable to describe as well as further focused efforts are being made on some of these.
  • Example CD123 and CD70 are the prime targets taken into consideration for “AND” gate technology. We at our end are focused on CD33/CD123 combination on our upcoming communication. In fact, CD70 has been also taken into consideration. In recent work from Prof. Maus lab from MGH this work was presented in ASH meetings.
  • We do not think adding recent information takes away the impact of this review. We are not competing with any other manuscript. Our focus is to bring what is happening in the field and modulation that is taking place in some of the prime labs in the world to address the challenge of heterogeneity of AML and maximize targeting efficiency and minimize off-target toxicities, including HSCs.
  • We have also elaborately explained the modifications that could take up in the clinical front, as these studies in preclinical models have shown such remarkable results.

Comments Addressed

Lines 55-56: “ … provide a T cell with new antigen specificity.” -> “… provide a T cell with an alternative antigen specificity.” is more correct regarding that T cells keep their own TCR. Only, CRISPR CAS9 approach leading to endogenous TCR KO leads to a new antigen specificity.

  • We have included this statement.

Line 58: “Major Histocompatibility Complex (MH

Line 60-62: “This resistance is avoided due to the functional nature of the MHC-independent to the functional nature of the MHC-independent mode of action of CAR-T cells”: correct the sentence

  • this sentence has been rectified.

Line 62-63: “ T cell receptors (TCR)-ζ “ Zeta chain do not exist in the TCR. However, there is a zeta chain among the CD3. Thus, CARs incorporating CD3zeta signaling domain.

  • We have made these changes, we has also established in our CAR signaling.

Lines 68 – 76: Authors resume the method of CAR-T therapy. However, an important step is missing. Patients are conditioned before CAR-T cells infusion in most of the cases. Please add this step.

  • In the revised version of the manuscript, we have included these changes

Line 613: “anitgens” please correct

  • Yes, rectification has been made

Line  631 – 634: “NOT” need to be more explained and add clearly in the related figure.

We would like to request more information.

  • Would the reviewer like us to add a figure or further in-depth explanation would suffice?
  •  

References:

We have included all the references. We had some trouble initially as the references were getting duplicated, but for now, we have included the references as well as added some more to support of statements. We in this manuscript have described the author's name including the references and then further described the efforts with preclinical work in those references.

Line 533; Novartis CAR trial with CTL109, authors have to add the trial number (NCT).

  • We have now included the NCT nos

Line 586: “Detailed reviews that summarize all these factors are readily available in literature.”, authors have to add references for some of these detailed reviews.

  • We have additional added three references to support his statement.

Figures:

Nothing is in place at least in my manuscript version, numbers and legends are missing.

  • We apologize for this. The effort on our end is to not distort the outline of the manuscript as intended by the journal. We are making sure that the figures are in one place and their figure legend is mostly followed on the second page. We are having difficulty as the figure legends are in a box with doesn’t take “line nos”

The figure 3 is cut in 3 parts (?).

  • Yes it got cut while sending the figure legend. We have now fixed that to make it more elaborate for explanation as well as easy to understand.
  • Our goal to add cartons to circumcise the explanation on our part. We don’t want to copy from any other review of format. We and our team which support producing figures of or all these papers have kept in our mind to focus and portray our own idea. Example: Weber’s work which recently came out in a science journal from Prof. Mackall’s lab has shed the light of reversing the exhausted phenotype of CAR-T cells which were chronically exposed to tumor antigen and/or tonic signaling. This was considered as rejuvenation of CAR-T clones with epigenetic remolding, in short the CAR-T cells which were induced to have a brief rest period showed better anti-tumor capacity. We have depicted this in our manuscript by an icon of “COFFEE cup”. These are our innovative ideas to bring out our ways to explain readers in a nutshell context.

Authors must organize clearly figures and legends in the manuscript. The Figure numbers need to be add in the text.

  • We have addressed these comments. Our figure 2 has been labeled now been individually labelled and supported in figure legends.

In addition, the text in the figures is too small; authors have to increase the size.

  • We are open to your ideas. The journal has it’s a pattern where they have suggested using 10 as font size and with the font “Palatino Linotype”

Reviewer 3 Report

This manuscript reviews aspects of current targeting of AML with CAR-T technology. The manuscript is in general well written, contains no obvious misinformation and the references are adequate. This reviewer has the following, very minor suggestions:

  • The abstract is too vague. More specific statements/conclusions of the review with regards to AML and targets should be made. Directions of further research and clinical application must be stated.
  • The introductory lymphoma part is well accompanied by schematic Figures, whereas the following AML part is somewhat "long"; I wonder whether some illustration / additional Figure is possible to facilitate the reading.
  • CAR-T in AML is characterzed by ... inexistence in the clinics, in contrast to DLBCL/ALL. This should be discussed with more emphasis; why ist that? where are the real problems ? is AML really a field for CAR-T ?

Author Response

Hi Reviewers,

Thank you for your esteem time and efforts to provide us your valuable inputs. We have addressed most of the comments and looking forward to hearing from you. Our efforts as I have explained as well is to bring out the lasted in the field of CAR- T cells redirected towards AML. We are well aware that there is pleotropic list of targets and even more coming out as well write and communicate. This is our humble effort to bring out the current preclinical efforts that has been successfully found their way in jounrals and publications. The list is exhaustive so we have clear mindset of provding the “Emerging CAR T cell strategies for the treatment of AML”. We not only focus on some challenging targets but have in depth provided ways in which efforts are put together by many labs to putatively target more than one tumor antigen to increase the specificity of the CAR-T cells.

We would like to know which other targets would reviewer like us to describe and contribute further.

  • We would take all suggestions into consideration. We found few of these targets more amicable to describe as well as further focused efforts are being made on some of these.
  • Example CD123 and CD70 are the prime targets taken into consideration for “AND” gate technology. We at our end are focused on CD33/CD123 combination on our upcoming communication. In fact, CD70 has been also taken into consideration. In recent work from Prof. Maus lab from MGH this work was presented in ASH meetings.
  • We do not think adding recent information takes away the impact of this review. We are not competing with any other manuscript. Our focus is to bring what is happen in the field and modulation that are taking place in some of the prime labs in the world to address the challenge of heterogeneity of AML and maximize targeting efficiency and minimizing off target toxicities, including HSCs.
  • We have also elaborately explained the modifications that could take up in the clinical front, as these studies in preclinical models have shown such remarkable results.

Comments Addressed

Lines 55-56: “ … provide a T cell with new antigen specificity.” -> “… provide a T cell with an alternative antigen specificity.” is more correct regarding that T cells keep their own TCR. Only, CRISPR CAS9 approach leading to endogenous TCR KO leads to a new antigen specificity.

  • We have included this statement.

Line 58: “Major Histocompatibility Complex (MHC)”

  •  

Line 60-62: “This resistance is avoided due to the functional nature of the MHC-independent to the functional nature of the MHC-independent mode of action of CAR-T cells”: correct the sentence

  • this sentence has been rectified.

Line 62-63: “ T cell receptors (TCR)-ζ “ Zeta chain do not exist in the TCR. However, there is a zeta chain among the CD3. Thus, CARs incorporating CD3zeta signaling domain.

  • We have made these changes, we has also established in our CAR signaling.

Lines 68 – 76: Authors resume the method of CAR-T therapy. However, an important step is missing. Patients are conditioned before CAR-T cells infusion in most of cases. Please add this step.

  • In the revised version of the manuscript we have included these changes

Line 613: “anitgens” please correct

  • Yes, rectification has been made

Line  631 – 634: “NOT” need to be more explain and add clearly in the related figure.

We would like to request for more information.

  • Does the reviewer would like us to add a figure or further in-depth explanation would suffice?
  •  

References:

We have included all the references. We had some trouble initially as the references were getting duplicated, but for now we have included the references as well as add some more to support of statements. We in this manuscript have descried the authors name including the references and then further described the efforts with preclinical work in those references.

Line 533; Novartis CAR trial with CTL109, authors have to add the trial number (NCT).

  • We have now included the NCT nos

Line 586: “Detailed reviews that summarize all these factors are readily available in literature.”, authors have to add references for some of these detailed reviews.

  • We have additional added three references to support his statement.

Figures:

Nothing is in place at least in my manuscript version, numbers and legends are missing.

  • We apologies for this. The efforts on our end is to not distort the outline of the manuscript as intended by the journal. We are making sure that the figures are in one place and their figure legend is mostly followed on second page. We are having difficult as the figure legends are in a box with doesn’t take “line nos”

The figure 3 is cut in 3 parts (?).

  • Yes it got cut while sending the figure legend. We have now fixed that to make it more elaborate for explanation as well as easy to understand.
  • Our goal to add cartons to circumcise the explanation on our part. We don’t want to copy from any other review of format. We and our team which support producing figure of or all these papers have kept in our mind to focus and portray our own idea. Example : Weber’s work which recently came out in science journal from Prof. Mackall’s lab has shed light of reversing the exhausted phenotype of CAR-T cells which were chronically exposed to tumor antigen and/or tonic signaling. This was considered as rejuvenation of CAR-T clones with epigenetic remolding, in short the CAR-T cells which were induced to have a brief rest period showed better anti-tumor capacity. We have depicted this in our manuscript by an icon of “COFFEE cup”. These are our innovated ideas to bring out our ways to explain reader in a nut shell context.

Authors must organize clearly figures and legends in the manuscript. The Figure numbers need to be add in the text.

  • We have addressed these comments. Our figure 2 has been labeled now been individually labelled and supported in figure legends.

In addition, the text in the figures is too small; authors have to increase the size.

  • We are open to your ideas. The journal has it’s pattern where they have suggested using 10 as font size and with font “Palatino Linotype”

Round 2

Reviewer 2 Report

The authors have addressed most of the comments and the manuscript has drastically gain in quality. The new title is, from my point of view, much more related to the work and the manuscript is now much more clear and interesting.

Line  631 – 634: “NOT” need to be more explained and add clearly in the related figure.

We would like to request more information.

  • Would the reviewer like us to add a figure or further in-depth explanation would suffice?

Addition in Figure 2 of a “NOT” panel would be nice but regarding the present size of the Figure 2, I think that the description in the text is enough to avoid an unclear figure 2.

The minors comments has to take in consideration:

  • The Figure numbers are still not included in the text! We do not know when we have to look at the figure when we are reading the text. Like all original articles or reviews, authors MUST include in the text “(Figure X)” in some related sentences.
  • In the legend of the Figure 2, H and G have to be reverse in order to have classical A, B, C, D, E, F, G and H.
  • Text from line 634 to 649 has not to be in italic but has to be justified.
  • In my pdf version, the bottom of the figure 3 is cut, we don’t have all the text for the section “Modulation of Tumor Microenvironnement and Extensible”

Author Response

Hi Reviwer.

Thank you for your positive feedback.

Comments 

Addition in Figure 2 of a “NOT” panel would be nice but regarding the present size of the Figure 2, I think that the description in the text is enough to avoid an unclear figure 2.

  • We have added in section 3.8 one another target with the proof of concept explaining the NOT gate strategy

The Figure numbers are still not included in the text! We do not know when we have to look at the figure when we are reading the text. Like all original articles or reviews, authors MUST include in the text “(Figure X)” in some related sentences.

In the legend of the Figure 2, H and G have to be reverse in order to have classical A, B, C, D, E, F, G and H.

  •  - We have Addressed these comments to the best of our knowledge. We have added the subtitle with  figure nos
  • We have addressed figure 2 and reversed the position of G and H

In my pdf version, the bottom of the figure 3 is cut, we don’t have all the text for the section “Modulation of Tumor Microenvironnement and Extensible”

- I really you receive the correct version this time. We address your comment in revision nos 3 and this is revision nos 4. We have added figure legends in a text box to minimize text fragmentation
